# EnsDiff: Ensemble Precipitation Nowcasting with Diffusion

## Abstract

Operational Numerical Weather Prediction (NWP) precipitation nowcasting usually considers forecast reliability by utilizing an ensemble of model forecasts. Existing data-driven methods often optimize MSE deterministically or resort to probabilistic forecasting with generative models. However, they only emphasize the optimization of the point forecast metrics, which makes it challenging to balance the trade-off between the optimization of accuracy and uncertainty. Human experts can hardly make an appropriate decision with an ensemble forecast when forecast calibration and sharpness are not considered. In this paper, we propose EnsDiff, which models the probability distribution to produce ensemble diffusion predictions. Not only does it outperform the existing models on a proper scoring rule, Continuous Ranked Probability Score (CRPS), but it also outperforms others on the deterministic metrics. Extensive experiments show that EnsDiff can enhance probabilistic, deterministic skills, and perceptual quality, outperforming state-of-the-art models.

## 1 Introduction

Heavy precipitation is one of the extreme events that can cause severe damage, such as flooding, in urban areas. Extreme rainfall can evolve from hours to minutes. Thus, an alert system aided by accurate and timely weather forecasts is crucial for the public to react to such high-risk events. Our study focuses on this precipitation nowcasting problem, where a high-resolution prediction of up to two hours ahead (Schmid et al., 2019) is required to inform the public to take the necessary actions for their safety. Traditional Numerical Weather Prediction (NWP), based on modeling the dynamics of atmospheric processes using differential equations, runs ensemble prediction to obtain state and uncertainty estimation that leads to significant computational and energy costs (Bauer et al., 2021). Thus, data-driven models that use abundant observation data to train can shorten the inference time of the model over the 0-2 hour lead time for precipitation nowcasting (Zhang et al., 2023).

Most data-driven approaches for precipitation nowcasting utilize radar imagery data available every several minutes and formulate radar prediction as a spatio-temporal deterministic forecast task. In particular, models are trained with some input frames to predict some future output frames that minimize losses such as the mean squared error (MSE). Although a deterministic forecast can be quickly computed to provide a good reference for the meteorologist, these precipitation field forecasts become blurry at longer lead times, corresponding to their uncertainty, which lacks representation of some small-scale convective patterns (Shi et al., 2015). On the other hand, ensemble forecast offers more economic value to decision makers than point forecast, when the model can recognize whether its forecast might be incorrect and express confidence when it is likely to be accurate (Zhu et al., 2002). Providing information about what might happen and how possible it is to happen allows a wide range of users, from emergency responders to everyday citizens, to take appropriate actions based on their risk tolerance and needs.

Although existing models produce realistic precipitation forecasts, they cannot yield high probabilistic skills when probabilistic forecasting has to maximize the sharpness of the predictive distributions, subject to calibration. Sharpness refers to how tightly the predictive distributions are concentrated, meaning that the distribution forecast is as precise as possible for people to refer to. Then, calibration describes the statistical alignment between probabilistic forecasts and actual observations. In essence, the observed outcomes should appear as random draws from the predicted distributions (Gneiting & Katzfuss, 2014). For example, a

forecast issues a 70% chance of rain. Then, it should rain approximately 70% of the time. A well-calibrated forecast enables decision-makers to adjust their decisions based on the forecast's confidence level. (Price et al., 2023). Existing models utilizing deterministic and probabilistic frameworks have poor ensemble calibration (Pathak et al., 2024). They train the forecast and diffusion models to optimize MSE and noise-prediction diffusion loss, which are not the objectives of optimizing calibration and sharpness. This is undesirable for modeling the uncertainty that aligns with the actual observations. Therefore, for the probabilistic precipitation nowcasting task, we propose a probabilistic model, that considers calibration and sharpness.

In this paper, we propose EnsDiff to produce ensemble forecasts with better calibration and sharpness by utilizing the equivalence of the Proper Scoring rule (Gneiting & Raftery, 2007) and the maximum likelihood estimation. Then, we utilize a deterministic model as a condition for the diffusion model to learn the forecast distribution effectively. It is observed that the deterministic model output the mean approximation when adding the mean prediction constraint. Conditioning on this mean prediction can enhance the deterministic skills of the final diffusion output, such as MSE and Critical Success Index (CSI), with a better location and shape precipitation prediction. EnsDiff enhances the sample quality while maintaining the probabilistic and deterministic skills by defining the task as a conditional forecasting task with the mean predictor being the condition model. This fully utilizes the mean predictor by increasing the likelihood of the samples conditioned on the mean predictor. Additionally, we propose to use EDM (Karras et al., 2022) in training a Latent Diffusion Model (LDM) (Rombach et al., 2022) that allows for a shorter sampling time, which favors the use of sampling ensembles.

The contribution of this work is summarized as follows:

- An ensemble forecasting diffusion model trained with the monotonic weighted diffusion objective corresponds to optimizing the ELBO of the output frame distribution with additive Gaussian noise. This can favor modeling the forecast distribution, which means better calibration and sharpness than merely emphasizing sample quality. Extensive studies have shown that the proposed method has beaten the SOTA results regarding the probabilistic skill - CRPS.

- We utilize a mean predictor model as a conditional model with an MSE constraint to guarantee its intermediate output quality. This, in turn, enhances the final diffusion model output in terms of deterministic skills.

- We propose incorporating classifier-free guidance to increase the likelihood of sampling under the mean predictor condition, which improves the sample quality by retaining the deterministic and probabilistic skill performance.

- To the best of our knowledge, we are the first to apply a continuous-time diffusion model on the precipitation nowcasting task that enables more efficient deployment of diffusion models in operational settings. Our model shortens the inference time and retains the probabilistic and deterministic performance on different datasets.

## 2 Related Work

We categorize precipitation nowcasting models into mean predictor models and generative models. The former uses mean squared error (MSE) as the loss, and the latter can learn the underlying distribution of training data to generate new samples, such as generative adversarial networks (GANs) and diffusion models.

### 2.1 Mean Predictor Models

Some earlier works (Shi et al., 2015; 2017; Wang et al., 2017) mainly adopt autoregressive methods that predict the output frames iteratively within a forecast window. However, one of the setbacks is that they optimize MSE loss, so it causes the model to tend to predict a blurry result (i.e., neighboring pixels have similar pixel values). Assume that we have $x$ as the input sequence and $y$ as the output sequence; the model is denoted as $M_\theta$ with parameters $\theta$. The model is trained to minimize the pixel-wise MSE loss, $\theta^* = \arg\min_\theta ||y - M_\theta(x)||_2^2$, to obtain the optimized $\theta^*$ parameters for inference. More blurriness in longer

lead-time prediction shows that the more uncertain the future frames are, the fewer details these models can predict. Meanwhile, it can be observed that the deterministic forecast is blurry, as if it is similar to the ensemble mean of an ensemble forecast. This shows the forecast uncertainty contributes to a blurry average state prediction (Price et al., 2023). In light of a similar magnitude of blurriness with the ensemble mean, we term these deterministic models "mean predictors".

These mean predictors adopt different architectural designs to strive for better modeling of the spatio-temporal dynamics. ConvLSTM (Shi et al., 2015) is the first to propose the integration of Convolutional layers to LSTM cells for modeling space and time relationships simultaneously. PredRNN (Wang et al., 2017) has a zigzag memory flow, when the memory is propagated to the horizontal and vertical stacked recurrent layers that can learn distributed representations capturing various aspects of the spatiotemporal dynamics. Earthformer (Gao et al., 2022b) applies cuboid attention to model local dynamics and global vectors to attend to these cuboids, which helps to represent the overall dynamics of the earth system.

## 2.2 Generative Models

With the deterministic models, though we can add noise to the initial input to run the model multiple times for an ensemble of forecasts (Sønderby et al., 2020), the resulting ensemble can only represent the initial condition uncertainty, but not the model uncertainty, as the model is deterministic. Also, the efficiency is low to run forward pass multiple times for the ensemble. Given the lack of model uncertainty representation and poor efficiency, it motivates more generative models widely applied in precipitation probabilistic nowcasting. In general, a generative framework predicts a distribution over possible future precipitation states, rather than a single deterministic forecast. This is achieved by modeling the conditional probability distribution with a generative model $P(X_{t+1:T} \mid X_{t-\tau:t})$ where $X_{t-\tau:t}$ represents the sequence of observed radar maps from time $t - \tau$ to $t$, $X_{t+1:T}$ denotes the sequence of future radar maps to be predicted from time $t + 1$ to $T$.

We introduce two main formulations for probabilistic forecasting with generative models: direct forecasting and residual forecasting.

### 2.2.1 Direct Forecasting

One of the first representative works is DGMR (Ravuri et al., 2021), which uses a conditional GAN conditioned on the input frames to generate the output frames, while regularization is also performed to minimize the differences between the generated and the true radar images, which can provide a highly skilled probabilistic forecast.

Another line of work using diffusion models offers a more stable training and mode coverage. Prediff (Gao et al., 2024) incorporates the Earthformer (Gao et al., 2022b) forecasting model into the UNet denoiser to denoise the output frames, allowing the denoiser to learn the mapping from the inputs to the output frames in the latent space. However, more parameters in the denoiser may be needed to correctly forecast and denoise the output frames simultaneously, and so it may converge slowly.

GenCast (Price et al., 2023) applies EDM (Karras et al., 2022) with the denoiser being GraphCast (Lam et al., 2023) on medium-range forecasting. The diffusion module models a different conditional distribution from other works, which is $p(y_{t+1} \mid y_t, y_{t-1})$ of the future state $y_{t+1}$ conditioning on the current and previous states. Then, it autoregressively samples from the conditional distribution to get the final joint distribution (i.e. $p(y_{1:T} \mid y_0, y_{-1}) = \prod_{t=0}^{T-1} p(y_{t+1} \mid y_t, y_{t-1})$). LDCast (Leinonen et al., 2023) learns a Latent Diffusion Model (LDM) that produces diverse predictions, having good calibration and sharpness. CasCast (Gong et al., 2024) similarly train a LDM, but it uses DiT (Peebles & Xie, 2023) to learn the cascaded representation of the precipitation that can successfully model both the deterministic part and stochastic components.

### 2.2.2 Residual Forecasting

It was shown that the prediction of a deterministic mean predictor model is similar to the mean of the ensemble from a diffusion model (Price et al., 2023). This observation is consistent with the models, in general, learning a deterministic regression to estimate the conditional mean and a stochastic diffusion model to correct the mean prediction by modeling the stochastic dynamics. DiffCast (Yu et al., 2024) carries

out an end-to-end training of the deterministic and stochastic diffusion models by constraining the global temporal consistency with an additional UNet. Once the regression model is trained, the residuals between the deterministic model output and the ground truth are learnt by the diffusion model. It outperforms some strong baselines, and has been proven to be a model-agnostic framework that can be applied to most of the SOTA mean predictors.

Stormcast (Pathak et al., 2024) has a similar framework to DiffCast, but instead they train the deterministic regression and stochastic diffusion in two stages. It needs more time to converge for each individual trained component, but the benefit is that the residual $t - 1$ conditioned by the diffusion is more accurate when it is trained in the first stage. And so, the residual $t$ denoised can be accurate. However, these methods have a poor calibration even though they can achieve competitive deterministic skills.

## 3 Methods

### 3.1 Task Formulation: Probabilistic Precipitation Forecasting

Precipitation nowcasting can be formulated as a spatio-temporal radar sequence forecasting task. Assume we can sample discrete-time sequences of images with length $M + K$ from a complete radar sequence that has observation data every several minutes in operation. The sequence can be segregated into $M$ input frames and $K$ sequentially following output frames, and each frame has dimensions $C \times H \times W$, corresponding to the number of channels, height, and width, respectively. In the case of a radar sequence forecasting task, the channel $C = 1$, as we have only one modality. Therefore, the task is to utilize the inputs $\mathbf{x} \in \mathbb{R}^{M \times C \times H \times W}$ to predict the outputs $\mathbf{y} \in \mathbb{R}^{K \times C \times H \times W}$.

Probabilistic forecasting refers to modeling the predictive conditional probability distribution $p(\mathbf{y}|\mathbf{x})$, where it aims to maximize the sharpness subject to calibration. When considering a probabilistic distributional forecast, the prediction space is $[F(\mathbf{y}), \mathbf{y}]$, where $F(\cdot)$ is the cumulative distribution function (CDF) of $p(\mathbf{y}|\mathbf{x})$, and $\mathbf{y}$ is the observed value. We can represent the cumulative distribution function as a probability integral transform random variable, $Z_F = F(\mathbf{y})$ (Gneiting & Katzfuss, 2014). Then, the distribution is said to be well calibrated when $Z_F \sim Uniform(0, 1)$ is close to a uniform distribution, indicating that the observation is indistinguishable from the ensemble members which forms the forecast distribution.

On the other hand, sharpness corresponds to the concentration of the predictive distribution $p(\mathbf{y}|\mathbf{x})$, when it does not depend on observation $\mathbf{y}$ (Gneiting & Katzfuss, 2014). As our task is to predict a real-valued precipitation intensity, we opt to minimize the mean width of the prediction interval, meaning the sharpness is maximized. Then, the ultimate goal of a probabilistic forecasting task is to constrain this minimization subject to the calibration matching the observation data distribution. Ensembles are the sample estimate of the full distribution by sampling from it to utilize this learned predictive distribution.

### 3.2 EnsDiff

We propose EnsDiff, which is an LDM (Rombach et al., 2022) that targets ensemble quality. First, we introduce the model architecture and training. Then, we discuss how to effectively model the forecast distribution with the Evidence Lower Bound (ELBO) objective, and thus the use of the EDM-monotonic objective. Finally, we further investigate the conditional generation for ensemble forecasting.

#### 3.2.1 Model Architecture and Training

With an LDM (Rombach et al., 2022) formulation, we first train a variational autoencoder (VAE) with 3D convolution layers. We label the encoder $\mathcal{E}(\mathbf{x})$ and decoder $\mathcal{D}(\mathbf{x})$. $\mathcal{E}$ projects the radar frames $\mathbf{x}$ to the latent space, where the mean predictor and diffusion model are trained. We apply SimVP (Gao et al., 2022a) as the mean predictor $\Psi$ to get the intermediate output frames in latent space, $\Psi(\mathcal{E}(\mathbf{x}))$. Then, EDM is adopted to learn the probabilistic distribution conditioned on the predicted output frames $\Psi(\mathcal{E}(\mathbf{x}))$ (or predicted mean). They are trained end-to-end with the EDM-monotonic objective. We explain in Section 3.2.3 the necessity of adding an MSE loss to constrain the mean predictor. The training pipeline is illustrated in Figure 1.

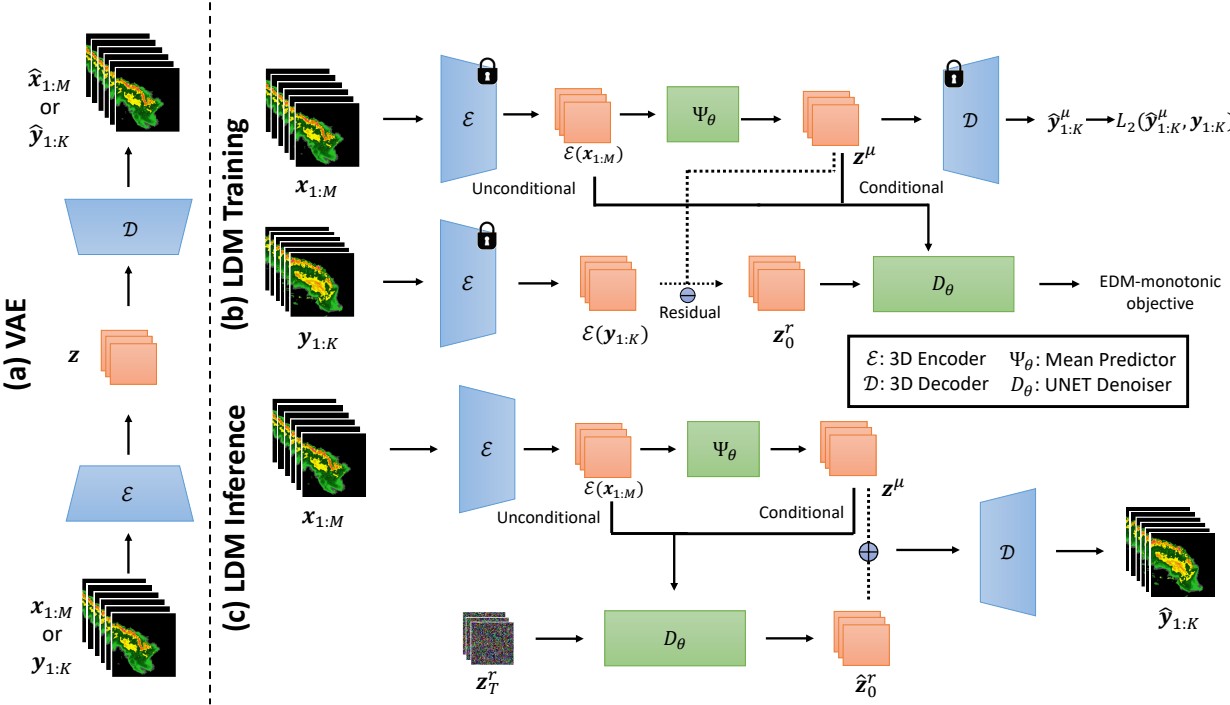

Figure 1: Overview of the EnsDiff pipeline: (a) illustrates the VAE training: $\mathcal{E}$ and $\mathcal{D}$ are first trained to learn the latent $\mathbf{z}$; (b) shows the LDM training: $\Psi_\theta$ is trained with the MSE constraint $L_2$. Then, classifier-free guidance has $\mathcal{E}(\mathbf{x}_{1:M})$ and $\mathbf{z}^\mu$ (i.e. $\Psi_\theta(\mathcal{E}(\mathbf{x}_{1:M}))$) as the unconditional and conditional signal respectively. The true residual $\mathbf{z}_0^r$ is computed by $\mathcal{E}(\mathbf{x}_{1:M}) - \mathbf{z}^\mu$. $D_\theta$ is trained with the EDM-monotonic objective by denoising $\mathbf{z}_0^r$; (c) shows the inference workflow: The input sequence, $\mathbf{x}_{1:M}$ is encoded by the Encoder, $\mathcal{E}$ into $\mathcal{E}(\mathbf{x}_{1:M})$. The mean predictor, $\Psi_\theta$ predicts the conditional latent variable, $\mathbf{z}^\mu$, while the denoiser UNet, $D_\theta$ predicts the residual, $\hat{\mathbf{z}}_0^r$ from Gaussian Noise, $\mathbf{z}_T^r$ with the classifier-free guidance. Then, $\mathcal{D}$ decodes to get the final prediction $\hat{\mathbf{y}}_{1:K}$;

Finally, during the sampling phase, we can obtain the ensemble forecast by sampling the diffusion model multiple times.

### 3.2.2 Probabilistic Forecasting with Diffusion

To utilize diffusion in probabilistic forecasting, we first understand the equivalence between diffusion objective and proper scoring rule (Gneiting & Raftery, 2007). Logarithmic score $\text{LS}(p_\theta(Y), y) = -\log p_\theta(y)$ is a local proper scoring rule (Parry et al., 2012), where $p_\theta(Y)$ is the density forecast of the weather random variable $Y$ following the unknown true distribution $p$, and $y$ is the observed value. Logarithmic score gives the negative log probability of the observed sample. Therefore, if we treat this score as a loss, we can minimize

$$L = -\mathbb{E}_{y \sim p}[\log p_\theta(y)] \tag{1}$$

This is equivalent to maximizing log-likelihood that can be derived from maximum likelihood estimation (Shao et al., 2024). Therefore, maximum likelihood estimation is a proper scoring rule, estimating parameters that minimizes the difference between the model $p_\theta(\mathbf{y})$ and true data $p(\mathbf{y})$ distribution.

Existing latent-diffusion forecasting models (Gao et al., 2024; Leinonen et al., 2023) learns with the simple diffusion noise prediction objective (Ho et al., 2020)

$$L_{\text{LDM}} = \mathbb{E}_{x,y,t,\epsilon \sim \mathcal{N}(0,\mathbf{I})} \left[ \| \epsilon - \epsilon_\theta(z_t, t, z_{cond}) \|_2^2 \right] \tag{2}$$

where $\epsilon$ is random noise added to the clean sample, $t$ is the denoising step of the reverse process, $z_t$ are the noisy latent-space output frames at denoising step $t$. $z_{cond}$ is the latent-space condition, where the model uses the latent-space input frames $\mathcal{E}(\mathbf{x})$ (Gao et al., 2024), or the mean-predictor $\Psi_\theta(\mathcal{E}(\mathbf{x}))$ (Leinonen et al., 2023). $\epsilon_\theta$ is the denoiser and $\theta$ represents the trainable parameters of the networks.

The above noise prediction objective for diffusion is derived from training a VAE with the variational upper bound (negative ELBO).

$$-\log p_\theta(\mathbf{y}) \leq -\text{ELBO}(\mathbf{y}) = \underbrace{D_{KL}\left(q\left(\mathbf{z}_1 \mid \mathbf{y}\right) \| p\left(\mathbf{z}_1\right)\right)}_{\text{Prior loss}} + \underbrace{\mathbb{E}_{q(\mathbf{z}_0 \mid \mathbf{y})}\left[-\log p\left(\mathbf{y} \mid \mathbf{z}_0\right)\right]}_{\text{Reconstruction loss}} + \underbrace{\mathcal{L}_T(\mathbf{y})}_{\text{Diffusion loss}} \tag{3}$$

where

$$\mathcal{L}_T(\mathbf{y}) = D_{KL}\left(q\left(\mathbf{z}_{t,\dots,1} \mid \mathbf{y}\right) \| p\left(\mathbf{z}_{t,\dots,1}\right)\right) \tag{4}$$

setting $\lambda = \log\left(\alpha_t^2/\sigma_t^2\right)$ to be the log signal-to-noise (SNR) ratio of diffusion timestep $t$, evidence lower bound (ELBO) of continuous-time diffusion models is simplified to (Kingma et al., 2021; Song et al., 2020):

$$-\text{ELBO}(\mathbf{y}) = \mathcal{L}_T(\mathbf{y}) + c = \frac{1}{2}\mathbb{E}_{t \sim \mathcal{U}(0,1), \boldsymbol{\epsilon} \sim \mathcal{N}(0,\mathbf{I})}\left[-\frac{d\lambda}{dt} \cdot \|\epsilon_{\boldsymbol{\theta}}\left(\mathbf{z}_t; \lambda_t\right) - \epsilon\|_2^2\right] + c \tag{5}$$

Since the loss weighting term is omitted from the simple diffusion loss (Equation 2)(Kingma et al., 2021), it does not equate to an ELBO objective. Thus, the simple diffusion objective favors sample quality over likelihood. While negative ELBO is an upper bound of the negative log-likelihood and negative log-likelihood is a proper scoring rule, the negative ELBO can be treated as another proper scoring rule. When $\text{LS}(p_\theta(Y), y) = -\log p_\theta(y)$, negative ELBO satisfies the proper scoring rule conditions (more details of the conditions in Appendix A.1.1)

$$E_P[S(P, Y)] \leq E_P[S(P_\theta, Y)] \tag{6}$$
$$-\log p(\mathbf{y}) \leq -\log p_\theta(\mathbf{y}) \leq -\text{ELBO}(\mathbf{y}) \tag{7}$$

Thereafter, the probabilistic forecasting distribution trained with the simple diffusion objective is unlikely to reconstruct the true data distribution.

To address this issue, a straightforward way is to multiply a weight in front of the simple diffusion objective (Kingma et al., 2021) with $-\frac{d\lambda}{dt}$, where $\lambda$ and $t$ are the log signal-to-noise (SNR) ratio and the diffusion step, respectively. We can denote the weight as $w(\lambda)$, as it depends on the $\lambda$. This methodology may trade too much sample quality for the likelihood, which may generate blurry results like the deterministic models, which is undesirable for forecaster reference.

We propose to use a monotonic decreasing function, EDM-monotonic (Kingma & Gao, 2024) that makes the diffusion objective as the ELBO on data augmented (i.e. additive noise) output frames. This objective simulates the modeling of some Gaussian noise-perturbed output frames, which forces the model not only reconstruct according to the observed output frames, but also allows the model to be robust enough to predict some perturbed outputs. Following the derivation on ELBO as a proper scoring rule, it means that the weighted diffusion objective represents a weighted proper scoring rule on different Gaussian noise-perturbed output frames. It can thus force the model to maximize the proximity to different noisy output frame distributions $P_\theta^t$ by minimizing the weighted proper scoring rule for different noisy diffusion timesteps $t$.

$$w(\lambda)S(P_\theta^t, z_t) = w(\lambda)E_{z_t \sim P^t}[S(P_\theta^t, z_t)] \tag{8}$$

By minimizing such a weighted scoring rule, the model can be more robust to different possible trajectories within a slight perturbation. The probabilistic forecast can achieve a nice calibration and sharpness, representing the uncertainty of the true observation while exploring the potential output trajectories. This has a similar benefit of data augmentation.

### 3.2.3 Diffusion Conditioned on Mean Predictor

From the perspective of a proper scoring rule, it is easier to learn the denoising forecast distribution with the mean statistics. Although the mean predictor has yet to be converged to the true mean, it acts as a useful information for learning the true forecast distribution $P$.

Similar to a downscaling task, learning the distribution of the residual between the predicted mean and ground truth is easier than learning that of the output frames directly due to a smaller variance in the residual (Mardani et al., 2023). The diffusion model can then denoise it with a more stable training. Thus, we adopt the residual forecasting framework with a mean predictor, and we condition the predicted mean to learn the output frame distribution with EDM.

Our deterministic and stochastic components are trained end-to-end. It is necessary to add a constraint to make the mean prediction closer to the ground truth, or else the intermediate mean prediction error as a condition will be propagated to the diffusion model.

**Mean prediction**  The first step involves learning a deterministic forecasting model, $\Psi_\theta(\cdot)$ on the latent to estimate the conditional mean:

$$\Psi_\theta(\mathcal{E}(\mathbf{x})) = \mathbb{E}[\mathcal{E}(\mathbf{y})|\mathcal{E}(\mathbf{x})] = \mu \tag{9}$$

This is achieved by training a gSTA module of SimVP (Cheng Tan & Li, 2022) as a mean predictor, $\Psi_\theta$ using paired data samples $\{x_i, y_i\}_{i=0}^N$ and optimizing the parameters $\theta$ with the Mean Squared Error (MSE) in the pixel space:

$$\min_\theta \frac{1}{N} \sum_{i=1}^N \|\mathbf{y} - \mathcal{D}[\Psi_\theta(\mathcal{E}(\mathbf{x}))]\|_2^2 \tag{10}$$

Squared error loss is a consistent scoring function to help converge to the mean of the forecast distribution (more details of consistent scoring function in Appendix A.1.2). This objective guarantees that the deterministic regression approximates the mean as close as possible. Thus, the intermediate mean prediction error as a condition propagated to the diffusion generation is minimized.

**Conditional Diffusion**  A diffusion model is trained to capture the conditional distribution $p(\mathbf{y} \mid \mu, \mathbf{x})$ We use $p(\mathbf{y_0})$ as the conditional distribution in the following formulations, omitting the conditions. Unlike other previous work (Gao et al., 2024; Yu et al., 2024; Gong et al., 2024), we utilize the continuous-time diffusion formulation (Song et al., 2020) since it allows more flexible and efficient sampling strategies. First, we have the forward process that adds i.i.d. Gaussian noise with variance $\sigma^2$ to the data distribution $p(\mathbf{y_0})$ and gets $p(\mathbf{y}; \sigma)$. Then, for large enough $\sigma_{\max}$, the distribution $p(\mathbf{y}; \sigma_{\max}^2) \approx \mathcal{N}(\mathbf{0}, \sigma_{\max}^2)$. Then, in the reverse process, it iteratively denoises towards $\sigma_0 = 0$. This denoising process can be implemented via the numerical simulation of the Probability Flow ODE (Song et al., 2020):

$$d\mathbf{y} = -\dot{\sigma}(t)\sigma(t)\nabla_{\mathbf{y}} \log p(\mathbf{y}; \sigma(t))dt \tag{11}$$

Here, $\nabla_{\mathbf{y}} \log p(\mathbf{y}; \sigma)$ is the score function (Hyvärinen & Dayan, 2005). Therefore, a model is trained to learn $\mathbf{s}_{\boldsymbol{\theta}}(\mathbf{y}; \sigma)$ that approximates the score function. This score function can be parameterized as $\nabla_{\mathbf{y}} \log p(\mathbf{y}; \sigma) \approx \mathbf{s}_{\boldsymbol{\theta}}(\mathbf{y}; \sigma) = \frac{D_{\boldsymbol{\theta}}(\mathbf{y};\sigma) - \mathbf{y}}{\sigma^2}$ (Ho et al., 2020), where $D_{\boldsymbol{\theta}}$ is a denoiser network that predicts the clean output frames $\mathbf{y}_0$. It is similar to minimizing the proper scoring rule with respect to different noise levels added to the clean data.

After the change of variable with $\lambda = -2\log\sigma$, the denoiser $D_{\boldsymbol{\theta}}$ is trained using the denoising score matching objective (Karras et al., 2022; Kingma & Gao, 2024):

$$\mathbb{E}_{(\mathbf{y_0},\mathbf{x})\sim p_{\text{data}}(\mathbf{y_0},\mathbf{x}),\lambda\sim p(\lambda)}[w(\lambda)\|D_{\boldsymbol{\theta}}(\mathbf{y}_\lambda; \lambda, \Psi_\theta(\mathcal{E}(\mathbf{x}))) - \mathbf{y}_0\|_2^2] \tag{12}$$

where $p(\lambda) = \mathcal{N}(\lambda; 2.4, 2.4^2)$ and $w(\lambda)$ is the weighting function. $\Psi_\theta(\mathcal{E}(\mathbf{x}))$ is the mean prediction conditional signal, and $\mathbf{y}_0$ is the clean output frames.

### 3.2.4   Mean Predictor-Free Guided Forecasting

EnsDiff uses classifier-free guidance (Ho & Salimans, 2022) to treat the mean prediction as a condition. It guides the denoising process towards the mean prediction conditioning signal. Then, we set the forecasting task as a conditional generation task based on the mean prediction, while the unconditional generation is still given the input frames to retain temporal consistency.

The unconditional model can be trained alongside the conditional one by randomly dropping the conditioning signal mean prediction $\Psi(\mathbf{x}_{1:M})$ (i.e. 15% of the time during training). Then, the unconditional signal is the input frames $\mathbf{x}_{1:M}$. This design allows the diffusion model to do direct and residual forecasting, which is predicting the output from input and mean prediction, respectively.

Then, during sampling, with the guidance scale $s \geq 0$ controlling the guidance strength, we calculate the predicted score function from

$$D_{\boldsymbol{\theta}}(\mathbf{y}; \sigma, \mathbf{x}) = sD_{\boldsymbol{\theta}}(\mathbf{y}; \sigma, \Psi(\mathbf{x}_{1:M})) - (s-1)D_{\boldsymbol{\theta}}(\mathbf{y}; \sigma, \mathbf{x}_{1:M}) \tag{13}$$

Thus, with updating the score, it decreases the unconditional likelihood of generating the output frames with direct forecasting while increasing the conditional likelihood of generating the output frames with residual forecasting. It reduces sample diversity in a way having a similar objective to increase the sharpness of the forecast distribution.

## 4   Experiments

### 4.1   Datasets

- **SEVIR** dataset (Veillette et al., 2020) covers 10,000 weather events in a 384km × 384km region in the US spanning a period of four hours with a 5-minute interval from 2017 to 2019. We extract the NEXRAD Vertically Integrated Liquid (VIL) data product from the five channels provided for precipitation nowcasting. Following previous works (Gao et al., 2022b; Seo et al., 2023), we predict the future VIL up to 60 minutes (12 frames) from 65 minutes of input frames (13 frames). We sample the test set from June 2019 to December 2019, leaving the remaining as the training set.

- **HKO-7** dataset (Shi et al., 2015) is a collection of radar echo data with 6-minute intervals, collected from 2009 to 2015 by the Hong Kong Observatory (HKO). It covers the region centered on Hong Kong with a radius of 256 km. We formulate the prediction task for up to the next 2 hours (20 frames) using a 30-minute observation period (5 frames). Data from 2009 to 2014 are used for training, while the test set consists of data collected in 2015.

Due to resource constraints, we configure the experiments on 128x128 resolution on all datasets by using bilinear interpolation to downsample the datasets from their original resolution. The dataset settings are summarized in Table 1.

Table 1: SEVIR and HKO-7 dataset settings

| Dataset | resolution | image size | interval | $L_{\text{in}}$ | $L_{\text{out}}$ |
|---------|-----------|-----------|----------|--------|---------|
| SEVIR   | 1 km      | 128       | 5 min    | 13     | 12      |
| HKO-7   | 2 km      | 128       | 6 min    | 5      | 20      |

### 4.2   Comparisons with State-of-the-Art Methods

To begin with, we observed that our proposed EnsDiff behaves differently on different datasets. Therefore, we argue that different components contribute to balancing the tradeoff between performance in different datasets. Thus, our proposed method is a more generalized framework compared to other SOTA models (more evaluation details in Appendix A.3). In addition, we only compared the inference time on the SEVIR and HKO-7 datasets, which is shown in Table 5. In general, the probabilistic models have multiple orders of magnitude longer inference time than the deterministic models. However, EnsDiff can manage to have a fast

Table 2: Performance comparison on SEVIR and HKO7 dataset. Both MAE and CRPS metrics are in the scale of $10^{-3}$. The best score among all models is highlighted in **bold**, while the best score among the probabilistic models, including ours, is underlined.

| Dataset | Model | Metrics | | | | | | | | |
|---|---|---|---|---|---|---|---|---|---|---|
| | | MAE↓ | SSIM↑ | LPIPS↓ | FVD↓ | CSI-m↑ | CSI$_4$-m↑ | CSI$_{16}$-m↑ | FSS-m↑ | CRPS↓ |
| SEVIR | ConvLSTM | 28.39 | 0.7216 | 0.3763 | 517.1 | 0.3458 | 0.3411 | 0.3607 | 0.6050 | - |
| | PredRNN | 28.91 | **0.7238** | 0.3572 | 528.8 | 0.3553 | 0.3702 | 0.4153 | 0.6286 | - |
| | SimVP | **27.75** | 0.7209 | 0.3673 | 413.1 | **0.3788** | 0.3803 | 0.4160 | 0.6452 | - |
| | Earthformer | 29.13 | 0.7102 | 0.3952 | 581.0 | 0.3556 | 0.3533 | 0.3838 | 0.6296 | - |
| | LDCast | 46.98 | 0.5772 | 0.3771 | 138.4 | 0.2193 | 0.2898 | 0.4598 | 0.4945 | 26.00 |
| | PreDiff | 41.44 | 0.6279 | **0.3047** | **35.40** | 0.3276 | 0.4271 | **0.6096** | 0.6483 | 30.34 |
| | DiffCast | 35.83 | 0.6410 | 0.3456 | 53.36 | 0.3305 | 0.4018 | 0.5504 | 0.6454 | 22.40 |
| | **EnsDiff** | 31.85 | 0.7117 | 0.3123 | 89.37 | 0.3602 | **0.4378** | 0.5927 | **0.6693** | **20.09** |
| HKO-7 | ConvLSTM | **40.11** | 0.5987 | 0.3679 | 768.9 | 0.2905 | 0.2628 | 0.2774 | 0.4084 | - |
| | PredRNN | 42.23 | 0.5785 | 0.3693 | 674.4 | 0.2857 | 0.2872 | 0.3263 | 0.4502 | - |
| | SimVP | 40.15 | 0.6039 | 0.3652 | 654.7 | 0.3020 | 0.2852 | 0.3115 | 0.4280 | - |
| | Earthformer | 41.67 | 0.5864 | 0.3839 | 815.0 | 0.2817 | 0.2532 | 0.2704 | 0.4014 | - |
| | LDCast | 109.14 | 0.2639 | 0.5622 | 1106 | 0.0260 | 0.0779 | 0.2939 | 0.1000 | 66.56 |
| | PreDiff | 48.01 | 0.5922 | 0.2672 | 107.0 | 0.2799 | 0.3787 | 0.5081 | 0.5093 | 42.39 |
| | DiffCast | 42.69 | **0.6198** | **0.2364** | **27.78** | 0.3013 | 0.4084 | **0.6084** | **0.5974** | **25.29** |
| | **EnsDiff** | 42.32 | 0.6101 | 0.2450 | 116.3 | **0.3025** | **0.4139** | 0.6019 | 0.5863 | 26.93 |

inference time among the probabilistic models, which is slightly slower than LDCast in both datasets. We will show in the later parts that our model also outperforms LDCast significantly, which means our model is a more viable choice.

**SEVIR**   From a probabilistic perspective, Table 2 shows that EnsDiff achieves the lowest CRPS among all models. Compared to other diffusion models, it has a lower CRPS by 10 % even from the strongest baseline DiffCast. This observation aligns with the MAE which has a 11% lower CRPS than DiffCast, where CRPS degenerates to MAE when evaluating a point forecast. It proves that our model is more preferrable to apply on both probabilistic and point forecasting tasks. EnsDiff, LDCast, and DiffCast are models that consist of a mean predictor and conditional diffusion in either latent or pixel space. Compared to the baseline having a similar architecture, our proposed EnsDiff achieves significant improvement across all metrics.

Additionally, it is noteworthy that while all probabilistic models perform worse in terms of MAE and SSIM, our proposed EnsDiff eliminates this degradation, achieving results comparable to those deterministic models. Compared to the best results achieved by probabilistic baselines, EnsDiff improves both MAE and SSIM by 11%. Furthermore, EnsDiff demonstrates similar performance in perceptual metrics, which are LPIPS and FVD, where deterministic models struggle due to blurriness at longer lead time. This indicates that EnsDiff not only enhances pixel-wise score and structural fidelity but also attains good perceptual quality as other probabilistic models.

In terms of the skill score, EnsDiff is able to predict different rainfall events more accurately than other probabilistic models, especially with the spatial tolerances of 4 and 8 pixels. Compared to probabilistic baselines, EnsDiff achieves the best CSI-m score, which considers pixel-wise accuracy, with a marginal improvement of 9%. It even outperforms most deterministic models, except for SimVP, which is only 5% lower. Moreover, EnsDiff improves the CSI$_4$-m and FSS-m metrics, which account for the medium spatial deviation of 4 and 8 pixels, by 2.5% and 3.2% respectively. However, its performance on CSI$_{16}$-m, which tolerates a larger spatial deviation, is slightly 2.7% lower than the best baseline, PreDiff. In summary, EnsDiff demonstrates the highest accuracy at medium scales (pool sizes of 4 and 8), and delivers comparable performance for both pixel-wise CSI and large pool sizes of 16.

In terms of qualitative comparisons, EnsDiff reconstructs the first few frames the best, which means that the short-term motion is well-captured. Interestingly, the accurate forecast can extend to a longer lead time where other models cannot, proving that temporal consistency and long temporal dependency are emphasized. In addition, from Figure 2, EnsDiff reconstruct the location and shape of the precipitation

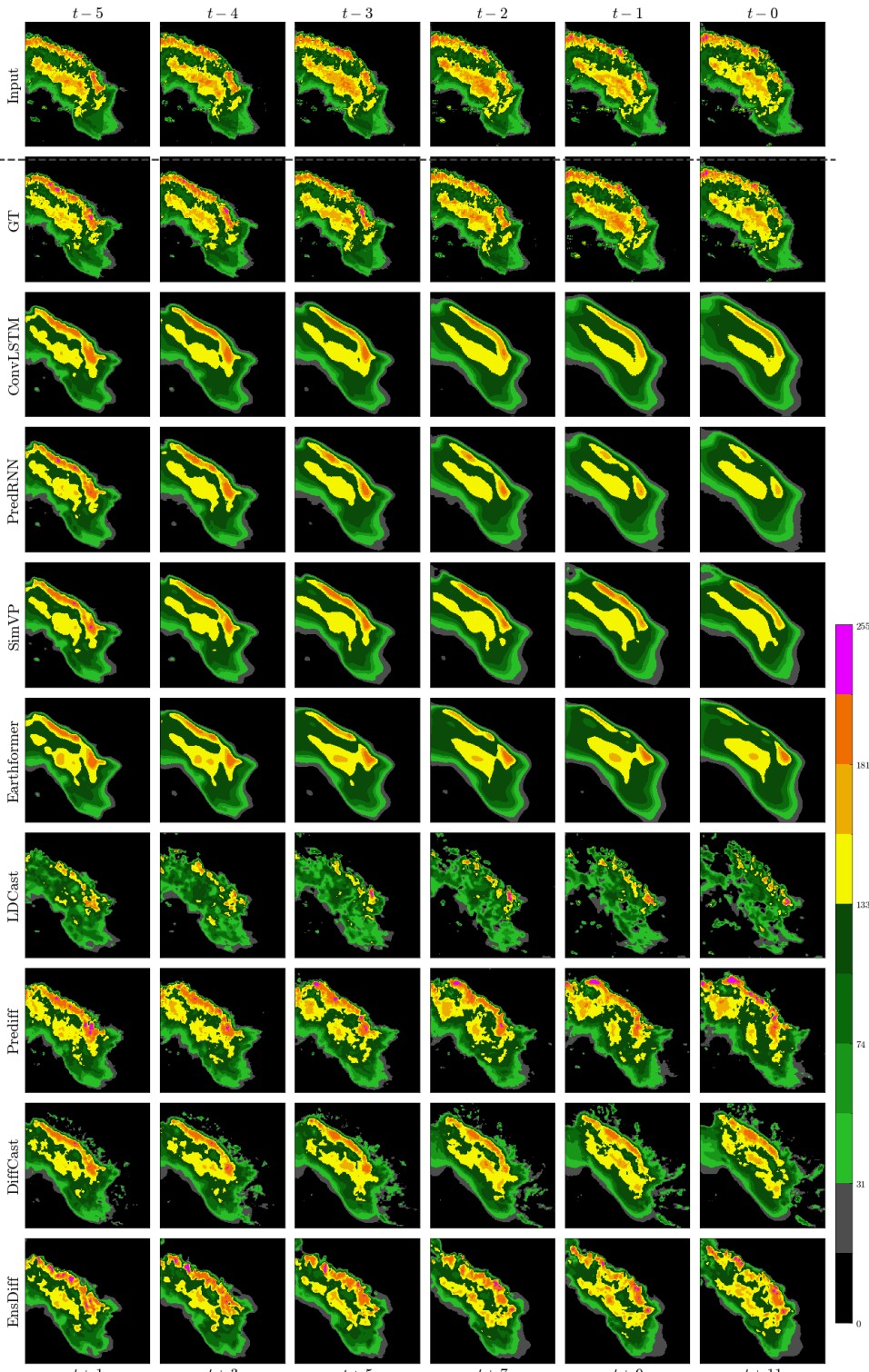

Figure 2: Visual results by state-of-the-art methods on samples from the SEVIR dataset.

cloud better compared to LDCast and DiffCast, where they underestimated some areas of the yellow regions. From Figure 5, it is obvious to see that our model can capture higher intensity precipitation from the shorter to the longer lead time.

**HKO-7**  In CSI and CSI-pool $4 \times 4$, EnsDiff outperforms all the baselines. This signifies that it better captures the formation and movement of precipitation with different thresholds in $1 \times 1$ and $4 \times 4$ scale. The two visualizations (Figures 7 and 8) clearly show that the shape and location are predicted correctly compared to other models.

However, it is observed that DiffCast, a pixel diffusion model, outperforms other latent diffusion models (LDM) with a huge gap in perceptual quality by looking at FVD, LPIPS, and SSIM. Meanwhile, EnsDiff has the smallest difference on the perceptual metrics with DiffCast. HKO-7 is a noisy dataset with scattered clouds. If a LDM projects these images to the latent, the scattered cloud has a high chance of being smoothed in the latent. Then, the LDM is trained with a less noisy version in the latent. Then, the prediction samples projected back to the pixel space would indeed lose details of those scattered behaviours. It results in a smoother or blurrier version in longer lead time, as shown in Figure 7 (during earlier training iterations). This lowers the sample quality as it cannot represent the scattered details similar to the ground truth.

With the loss of scattered details observation, the MAE and CRPS may result in a higher value. Despite that, by training for a sufficient of iterations, EnsDiff can lower the MAE and MSE to be smaller than all other probabilistic models (including DiffCast), and achieve a comparable CRPS with DiffCast (visualizations in Figures 6 and 8) Thus, we believe that LDM has to be trained for more iterations to achieve a higher pixel-wise accuracy MAE and probabilistic skill CRPS. EnsDiff has the best capability among all LDMs to overcome such sample quality degradation.

Interestingly, there is a significant tradeoff between CSI and other metrics on the HKO-7 dataset only. When we compare several checkpoints saved during training, we find that we can get better perceptual metrics (FVD, LPIPS, and SSIM) and probabilistic skill (CRPS) with a lowered CSI. For example, there is a model evaluated to have a higher CSI (0.3185) and CSI-pool $4 \times 4$ (0.4216) but a significantly higher CRPS (36.01). This is not desirable, so it requires more iterations of training to get a balance between CSI and CRPS. The tradeoff can be due to the loss of scattered details where the high-intensity areas are predicted in a patch shape, like in Figure 7. This kind of prediction can result in higher CSI, but lower in CRPS and other perceptual metrics.

## 4.3  Ablation Study

We study the effectiveness for each component contributing to the performance in the probabilistic and deterministic skills. Due to resource limitations, the ablation study is only performed on the SEVIR dataset. The result is reported in Table 3.

Table 3: Performance of different variants of EnsDiff on SEVIR dataset. The variants on the last row is the one reported in Table 2. Both MAE and CRPS metrics are in the scale of $10^{-3}$. CFG and CMP stand for classifier-free guidance and MSE-constraint applied on Mean Predictor, respectively. The best score is highlighted in **bold**, while the $2^{nd}$ best score is underlined.

| CMP | CFG | Mean Predictor | Monotonic Weighting | MAE↓ | SSIM↑ | LPIPS↓ | FVD↓ | CSI-m↑ | CSI$_4$-m↑ | CSI$_{16}$-m↑ | FSS-m↑ | CRPS↓ |
|---|---|---|---|---|---|---|---|---|---|---|---|---|
| ✓ | | gSTA | ✓ | **31.66** | 0.7112 | 0.3167 | 94.83 | 0.3603 | 0.4324 | 0.5775 | 0.6663 | **20.08** |
| | ✓ | gSTA | ✓ | 33.44 | 0.7051 | 0.3148 | 95.20 | 0.3481 | 0.4303 | 0.5967 | 0.6565 | 20.24 |
| ✓ | ✓ | AFNO | ✓ | 41.38 | 0.6715 | 0.3307 | 109.3 | 0.3129 | 0.3997 | 0.5662 | 0.5760 | 23.59 |
| ✓ | ✓ | gSTA | | 31.74 | 0.7108 | **0.3090** | **80.93** | **0.3703** | **0.4532** | **0.6138** | **0.6867** | 21.56 |
| ✓ | ✓ | gSTA | ✓ | 31.85 | **0.7117** | 0.3123 | 89.37 | 0.3602 | 0.4378 | 0.5927 | 0.6693 | 20.09 |

**Constraint on Mean Prediction**  Comparing the CFG and CMP+CFG variants, in most metrics (except CSI$_{16}$-m), CMP+CFG is better. This proves that adding a mean prediction constraint is crucial for the final forecast performance. There is a significant decrease in FVD which shows that the constraint can enhance the sample quality. Qualitatively, we observe that CFG has a poor long-range forecast capability in high intensity cases (Figure 9), where it has a similar prediction to ground truth in the first few frames. Meanwhile, the prediction is fair at most of the lead time in some smaller intensity cases (Figure 10). This can be reasoned by the fact that mean predictor can accurately forecast those low intensity precipitation with or without the constraint, while the constraint helps improve the mean prediction in those high intensity cases. Thus, the

final forecast from the diffusion can spend more its representation power in learning the residual that comes from the local stochastic dynamics.

**Mean Predictor-Free Guided Forecasting**  CMP and CMP+CFG variants have similar CRPS scores, and the FVD improves with the addition of CFG, which means that calibration and sharpness are not sacrificed when the sample quality is enhanced. Qualitatively, it is observed that each member of CMP+CFG aligns better with the ground truth, and so does the ensemble mean. The model without CFG tends to mispredict in some early frames $t + 3$, for example, the intensity of the yellow region is overpredicted to have a larger area in Figure 9, and the shape of the precipitation at $t + 3$ in the right-middle of Figure 10 is slightly dissimilar from the ground truth. The difference from ground truth is more obvious in a longer lead time. Thus, with CFG, EnsDiff achieves better temporal consistency and retains the capability to model the probabilistic distribution.

**Mean Predictor $\Psi_\theta$ model architecture**  In addition to the gSTA module (Cheng Tan & Li, 2022), we conducted an experiment where $\Psi_\theta$ is replaced with AFNO (Guibas et al., 2021), as suggested in LDCast (Leinonen et al., 2023). It is evident that using gSTA for $\Psi_\theta$ better captures spatiotemporal features in the latent space, resulting in the significant improvements across all metrics as shown in Table 3. The choice of the mean predictor is important, where it can be decided by comparing the performance of the mean prediction, like how we conduct experiment on the deterministic model. If we compare the general architecture between convolution (SimVP) and transformer (Earthformer) layers, SimVP outperforms Earthformer in all of the deterministic metrics. Thus, it is reasonable that EnsDiff using gsTA as a mean predictor outperforms that using AFNO.

**Monotonic Weighting**  We experiment with the gsTA without monotonic weighting. It is observed that its deterministic scores almost achieved the best among all variants but the probabilistic score CRPS has been reduced significantly. It proves that the model with monotonic weighting can indeed make the model train towards minimizing the proper scoring rule for different Gaussian-perturbed output. This in turn enhances the probabilistic skill of the ensemble forecast. Qualitatively, in Figure 9 and 10, comparing the variant CMP+CFG (gsTA) with and without monotonic weighting, it is observed that the one with the monotonic weighting has the precipitation predicted correctly in location and intensity from shorter to longer lead time.

## 5 Conclusion

To effectively apply a probabilistic forecasting model to precipitation nowcasting, the key is to have high probabilistic skills and retain high deterministic skills, perceptual quality, and inference speed. The monotonic weighting makes the diffusion objectives the ELBO on Gaussian-noised output frames, which helps gain calibration and sharpness. We frame the mean predictor as a condition for the diffusion model to learn the forecast distribution effectively with the mean approximation when deterministic skills are enhanced along with the mean prediction constraint. Then, we employ Mean Predictor-Free Guided Forecasting to fully utilize the mean predictor in a conditional forecasting way. Sample quality is improved when the likelihood or the probabilistic skill is not diminished.

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

# A  Appendix

## A.1  Additional Preliminary

### A.1.1  Proper Scoring Rule for Probabilistic Forecast

Let $P_\theta$ be the probabilistic forecast (a predictive distribution) provided for a random variable $Y$ following the unknown true distribution $P$, and let $y$ be the observed outcome. A scoring rule is a function:

$$S(P_\theta, y) = E_{Y \sim P}[S(P_\theta, Y)] \tag{14}$$

The score $S$ depends on both the forecast $P_\theta$ and the observed outcome $y$, which is the expected score computed under the true distribution $P$ of $Y$:

A scoring rule $S(P_\theta, y)$ is *proper* if the expected score is minimized when the forecast $P_\theta$ matches the true distribution $P$ for all possible forecasts $P_\theta$:

$$E_P[S(P, Y)] \leq E_P[S(P_\theta, Y)] \tag{15}$$

If equality holds only when $P_\theta = P$, the scoring rule is called *strictly proper*.

### A.1.2  Consistent Scoring Function for Point Forecast

Apart from obtaining a probabilistic forecast, practitioners sometimes prefer a point forecast for decision-making and easier communication (Gneiting & Katzfuss, 2014). When $x$ is the point forecast and $y$ is the observation, we compare competing model forecasts with a scoring function $s(x, y)$. In other words, the optimal point forecast is obtained by

$$\hat{x} = \arg \min_{x \in \mathbb{R}} \mathbb{E}_P[s(x, Y)] \tag{16}$$

when we minimize the expected scoring function $s$ under the distribution $P$ of the random variable future state $Y$.

As a special case of a proper scoring rule, a scoring function is *consistent* for a functional if the forecaster minimizes the expected score by forecasting the correct value of the functional, where a functional is a summary statistic derived from a probability distribution. (Gneiting & Katzfuss, 2014). Formally, let $P_\theta$ be a predictive distribution, $s(x, y)$ be the scoring function, and $T(P_\theta)$ be the functional of $P_\theta$ (e.g., the mean or median). The scoring function $s$ is consistent for $T$ if:

$$E_P[s(T(P_\theta), Y)] \leq E_P[s(x, Y)] \tag{17}$$

for all possible forecasts $x$. If equality holds only when $x = T(P_\theta)$, the scoring function is *strictly consistent*. A functional is elicitable if there exists a strictly consistent scoring function for it. In the precipitation nowcasting task, deterministic models commonly use mean square error (MSE) to train. The mean statistic of the distribution $P_\theta$ is elicitable with squared error loss, a strictly consistent scoring function. T

$$s(x, y) = (x - y)^2 \tag{18}$$

Thus, it is not surprising the deterministic forecast approximates the mean of the true distribution $P$. That is close to the ensemble mean of the predictive distribution $P$.

## A.2  Implementation Details

**SEVIR**  In general, SEVIR is an easier dataset than the other two, where there are clear patches and the movement or evolution of the cloud is more stable. Therefore, it does not need much special handling when applying EnsDiff on it. We utilize the reduced on plateau learning rate scheduler with early stopping. The model usually stops at epoch 50-60.

**HKO-7** Due to its geographical location in the Asia-Pacific, the observed rainfall is more scattered than in the SEVIR data, which has clear patches of precipitation. Therefore, when we apply diffusion to HKO-7, we must pay special attention to the latent shape and noise distribution.

The latent shape (height and width) plays a role when using our LDM for HKO-7. We experiment our model on the resolution of 128x128 and 256x256. We found that our model cannot converge on the 128x128 resolution data with latent shape 32x32 when the loss decreases, but the forecast is very poor that the precipitation intensity is always overestimated. However, the model trained on 256x256 with latent shape 64x64 can produce realistic forecast. Therefore, we speculate that EnsDiff is sensitive to the resolution and the detail of the signal. Insufficient information may cause the diffusion model difficult to converge. In view of this observation, we construct our model to have a larger latent height and width shape of 64x64 which can encapsulate more information from the original 128x128 signal.

For the noise distribution, the model is designed to be first trained with a noisier distribution and shifted to a less noisy distribution in a later stage. This echoes the training strategy that when the diffusion model starts training, we want the model to first learn the local stochasticities correctly, similar to the Numerical Weather Prediction (Bauer et al., 2015) when they discretize the grid to solve the partial differential equation and integrate over time. In fact, modeling the local dynamics is important since the chaotic behavior of the system (i.e., the sensitivity of the initial condition from a local point) may greatly affect the global dynamics in the long run. This is the exact opposite to training diffusion model on high-resolution natural images (like in ImageNet), where we first train on a smaller noise schedule with a smaller noise value, and then shift the noise schedule to a higher noise value (Hoogeboom et al., 2023).

Following EDM, our model has a noise distribution of $\log \sigma \sim \mathcal{N}\left(P_{\mathrm{mean}}, P_{\mathrm{std}}^2\right)$. $P_{\mathrm{mean}}$ (default is -1.2) can be used to control the distribution shifted to high or small noise. We set the standard deviation to be the default $P_{\mathrm{std}} = 1.2$ which gives a suitable range of noise levels. First, we start with a noise distribution in the higher noise region, where $P_{\mathrm{mean}} = 0$. The training is more stable where the noise can be denoise gradually throughout several epochs. The intensity will not be overestimated when compared to starting with the smaller noise region. After training for 50 epochs, it is shifted towards a noise distribution with smaller noise by setting $P_{mean} = -1.2$. And further trained with 50 more epochs, it is further shifted to a less noise region by setting $P_{mean} = -2.4$. With this design, our model can learn well from the local to global dynamics.

### A.3 Evaluation

Besides the diffusion-based probabilistic models (i.e. LDCast (Leinonen et al., 2023), PreDiff (Gao et al., 2024) and DiffCast (Yu et al., 2024)), we also compare the performance of our proposed EnsDiff with several deterministic models: ConvLSTM (Shi et al., 2015), PredRNN (Wang et al., 2017), SimVP (Gao et al., 2022a) and Earthformer (Gao et al., 2022b). All probabilistic models are evaluated across 10 ensemble members.

### A.3.1 Probabilistic Skills

To evaluate probabilistic skills, the Continuous Ranked Probability Score (CRPS) is one of the Proper Scoring Rules (Gneiting & Raftery, 2007), which is widely used by the weather community to assess the accuracy and uncertainty of probabilistic forecasts for continuous variables. It provides a penalty for both overconfident and underconfident forecasts, quantifying both calibration and sharpness. A lower CRPS indicates a better forecast, with 0 being perfect.

$$\mathrm{CRPS}(F, y) = \int_{-\infty}^{\infty} \left(F(x) - \mathbb{I}(x \geq y)\right)^2 \, dx \tag{19}$$

where $F(x)$ is the cumulative distribution function (CDF) of the forecasted distribution, $y$ is the observed value, $\mathbb{I}(x \geq y)$ is the indicator function, which equals 1 if $x \geq y$, otherwise 0. When evaluating a point forecast, it reduces to a Mean Absolute Error (MAE), where $\hat{y}$ is the predicted deterministic value.

$$\mathrm{CRPS} = |y - \hat{y}| \tag{20}$$

Also, rank distribution is used to evaluate how well the ensemble forecasts cover the observations. A uniform rank histogram implies good calibration, while U-shaped or domed histograms indicate issues with overconfidence or excessive uncertainty, respectively.

### A.3.2   Deterministic Skills

For deterministic skills, following previous works, Mean Absolute Error (MAE) and Structural Similarity Index (SSIM) are reported as the models' pixel-wise score and structural accuracy, respectively. In addition, the average Critical Success Index (CSI) measures the models' capability for predicting different rainfall events with the corresponding threshold values (Shi et al., 2015; Gao et al., 2022b). Aside from pixel-wise metrics, we also report the average CSI scores at the spatial pooling scales of $4 \times 4$ and $16 \times 16$, and the Fractional Skill Score (FSS) with a pool size of 8. These pooled metrics evaluate the models' accuracy within a local region rather than pixel-wise.

Additionally, LPIPS and Fréchet Video Distance (FVD) assess the visual quality of the model predictions. FVD measures the similarity between the learned data distribution and the true data distribution via encoded video latent features with a pre-trained model.

### A.4   In-depth Discussion

**Proper Scoring Rule: CRPS**   We study CRPS with spatial pooling by combining forecasts and observations before calculating CRPS. We applied the pooling on the scale of $4 \times 4$ and $16 \times 16$ pixels, similar to the CSI-pool. The CRPS with pooling reflects the average performance of the probabilistic forecast across the region. Table 4 shows the CRPS scores for all 3 datasets. For SEVIR, we can see that EnsDiff outperforms the other diffusion models in most of the CRPS with or without pooling. And for HKO-7, EnsDiff performs slightly worse than DiffCast. This shows that EnsDiff not only has accurate probabilistic distributed forecast pixel-wise, but it also captures the spatial patterns well in the scale of $4 \times 4$ and $16 \times 16$.

Table 4: CRPS comparison on SEVIR and HKO-7 dataset. All are in the scale of $10^{-3}$. The best score among all models is highlighted in **bold**, while the $2^{nd}$ best score is underlined.

| Dataset | Model | Metrics | | |
|---|---|---|---|---|
| | | CRPS↓ | CRPS$_4$↓ | CRPS$_{16}$↓ |
| SEVIR | LDCast | 26.00 | 22.80 | 17.98 |
| | PreDiff | 30.34 | 25.62 | 18.62 |
| | DiffCast | 22.40 | 17.98 | 11.09 |
| | **EnsDiff** | **20.09** | **16.24** | **10.24** |
| HKO-7 | LDCast | 66.56 | 64.07 | 61.74 |
| | PreDiff | 42.39 | 37.09 | 27.92 |
| | DiffCast | **25.29** | **20.04** | **12.68** |
| | **EnsDiff** | 26.93 | 21.91 | 14.74 |

**Ensemble mean comparison**   Apart from examining the quantitative probabilistic skill scores, it is useful to check the ensemble mean visualizations. In this way, we can verify whether the ensemble members can contribute to reconstructing the true forecast distribution.

Calibration refers to how well the predicted probabilities (or distributions) match the observed outcomes. A well-calibrated ensemble forecast means that the ensemble predictions are statistically consistent with the observations. The ensemble mean itself cannot fully assess calibration because calibration is about the entire distribution, not just its average. However, if the ensemble mean is systematically biased compared to the observed outcomes (e.g., consistently overpredicts or underpredicts), it indicates bias in the forecast, which is a component of miscalibration. For example, in SEVIR (Figure 3 and 4), we observe that Prediff has consistently higher intensity in its ensemble mean than the ground truth. Meanwhile, EnsDiff and DiffCast both have similar intensity values with the ground truth, which means they do not consist of a significant bias.

## B   Inference Time Comparison

Table 5: Inference time comparison on SEVIR and HKO7 dataset. All values are in seconds. '/' indicates it is out-of-memory for the model with the specific batch size. The best among all models is highlighted in **bold**, while the best among the probabilistic models, including ours, is underlined.

| Dataset | Model | Batch Size | | | | | |
|---|---|---|---|---|---|---|---|
| | | 1 | 2 | 4 | 8 | 16 | 32 |
| SEVIR | ConvLSTM | 0.1265 | 0.0695 | 0.0436 | 0.0302 | 0.0191 | 0.0171 |
| | PredRNN | 0.2608 | 0.1432 | 0.0868 | 0.0543 | 0.0406 | 0.0214 |
| | SimVP | **0.1114** | **0.0597** | **0.0333** | 0.0236 | 0.0239 | 0.0224 |
| | Earthformer | 0.1602 | 0.0651 | 0.0376 | **0.0209** | **0.0131** | **0.0094** |
| | LDCast | 4.4744 | 3.0824 | 2.3811 | 1.991 | 1.7998 | 1.7517 |
| | PreDiff | 73.0939 | 66.7454 | 60.5865 | 58.4567 | 56.6579 | / |
| | DiffCast | 13.9273 | 8.8283 | 12.1663 | 14.9587 | 11.7392 | 11.6339 |
| | **EnsDiff** | 3.7287 | 3.1714 | 2.7292 | 2.5782 | 2.4828 | 2.4304 |
| HKO-7 | ConvLSTM | 0.1231 | 0.0666 | 0.0427 | 0.0324 | 0.0203 | 0.0183 |
| | PredRNN | 0.2616 | 0.1536 | 0.0897 | 0.0547 | 0.0438 | 0.023 |
| | SimVP | 0.1247 | 0.0664 | 0.0403 | 0.0289 | 0.027 | 0.0274 |
| | Earthformer | **0.1187** | **0.0611** | **0.0333** | **0.0196** | **0.0134** | **0.0099** |
| | LDCast | 10.3323 | 3.415 | 2.7867 | 2.5037 | 2.3738 | 2.314 |
| | PreDiff | 70.797 | 65.4263 | 59.2361 | 59.2361 | 55.4093 | / |
| | DiffCast | 55.4445 | 33.8662 | 34.1664 | 45.9586 | 33.3385 | 31.4788 |
| | **EnsDiff** | 7.8582 | 7.5986 | 7.4277 | 7.3929 | 7.298 | / |

## C   More Qualitative Results

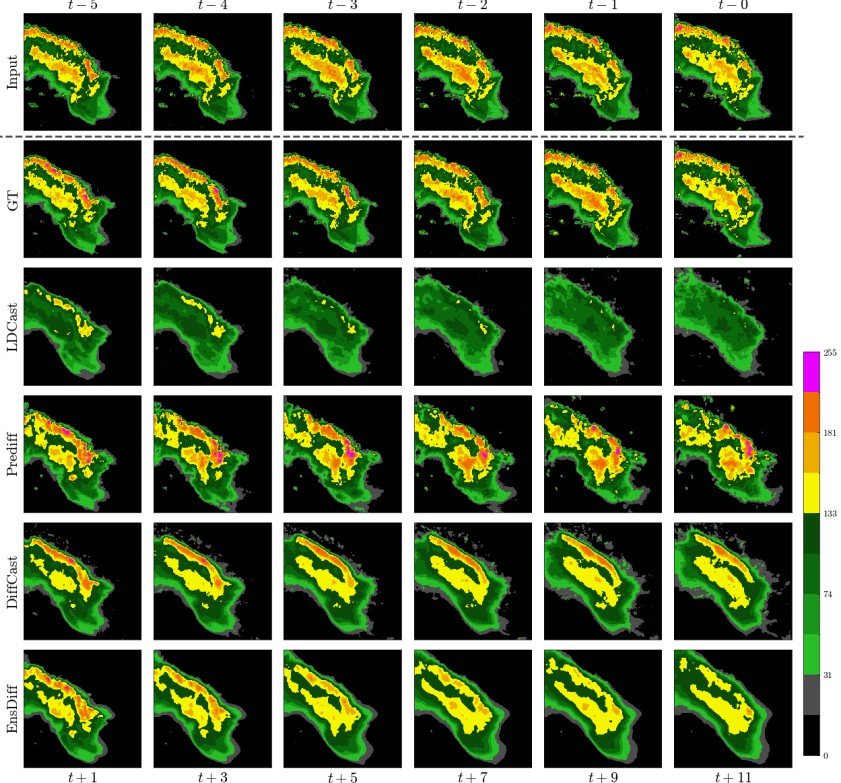

Figure 3: Visualization of the ensemble mean on the SEVIR dataset (Sample 1)

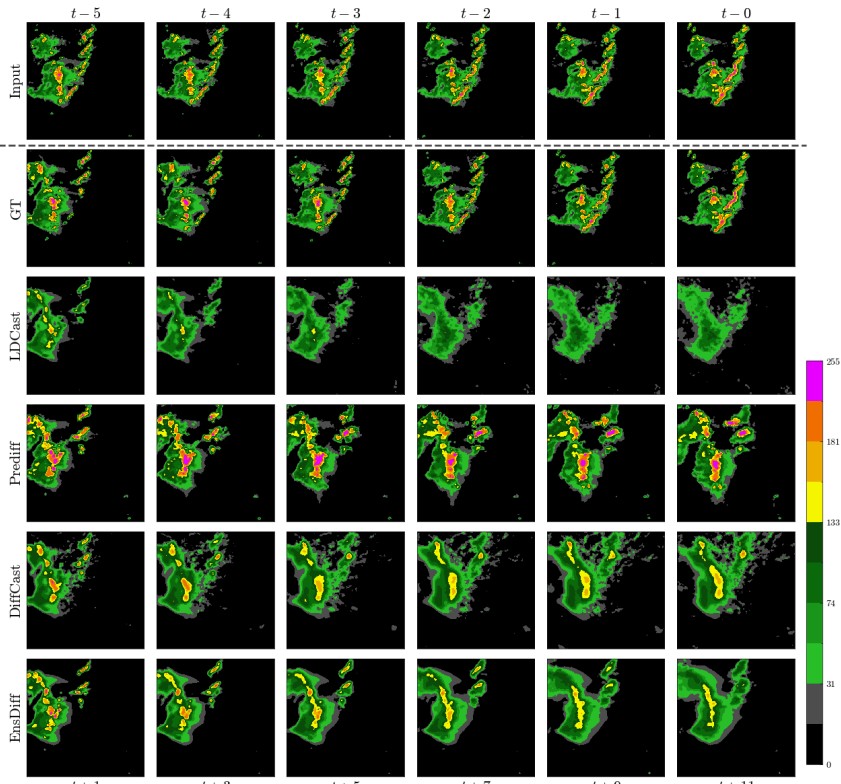

Figure 4: Visualization of the ensemble mean on the SEVIR dataset (Sample 2)

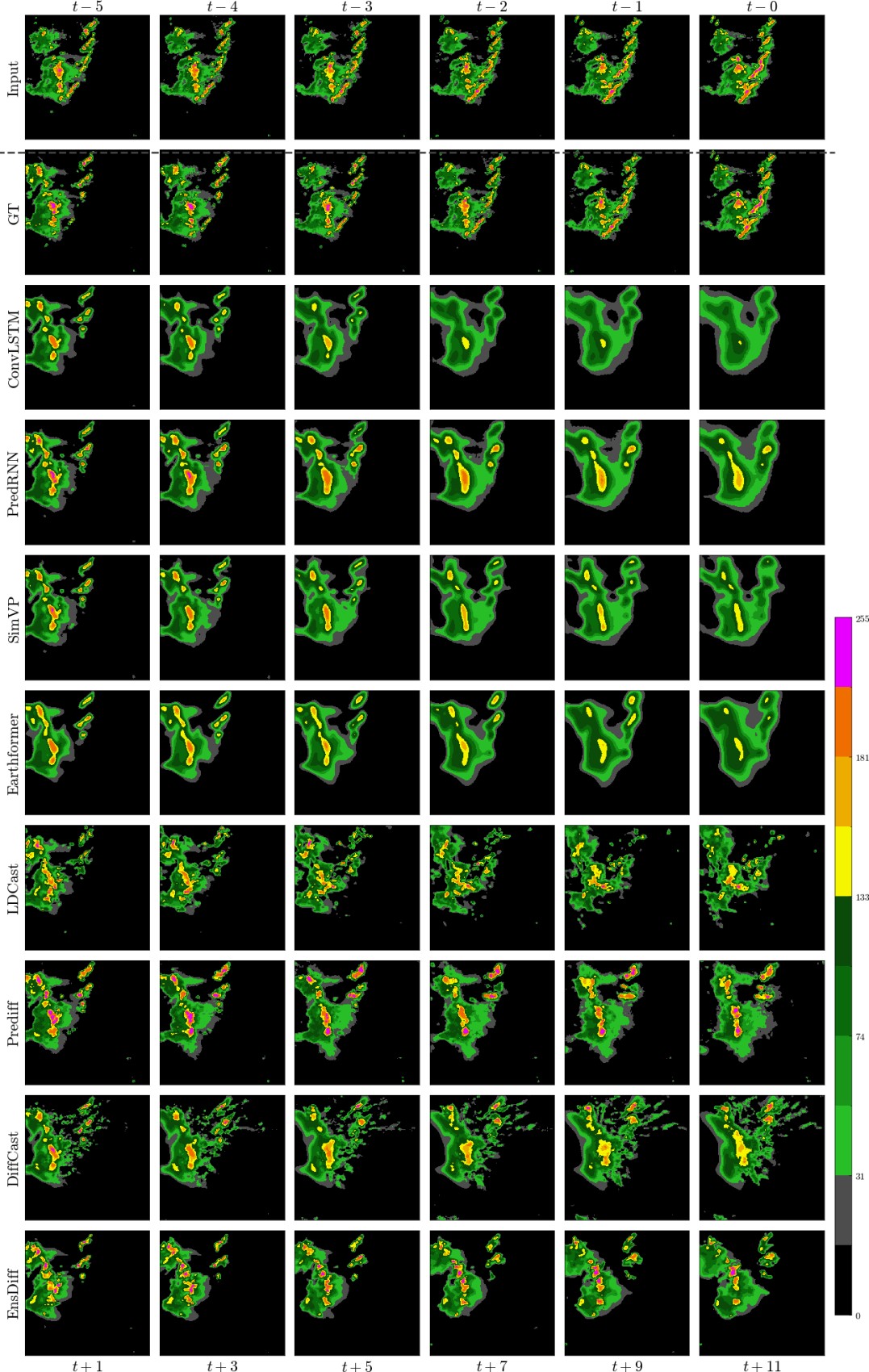

Figure 5: Visualization of all the models on the SEVIR dataset (Sample 2)

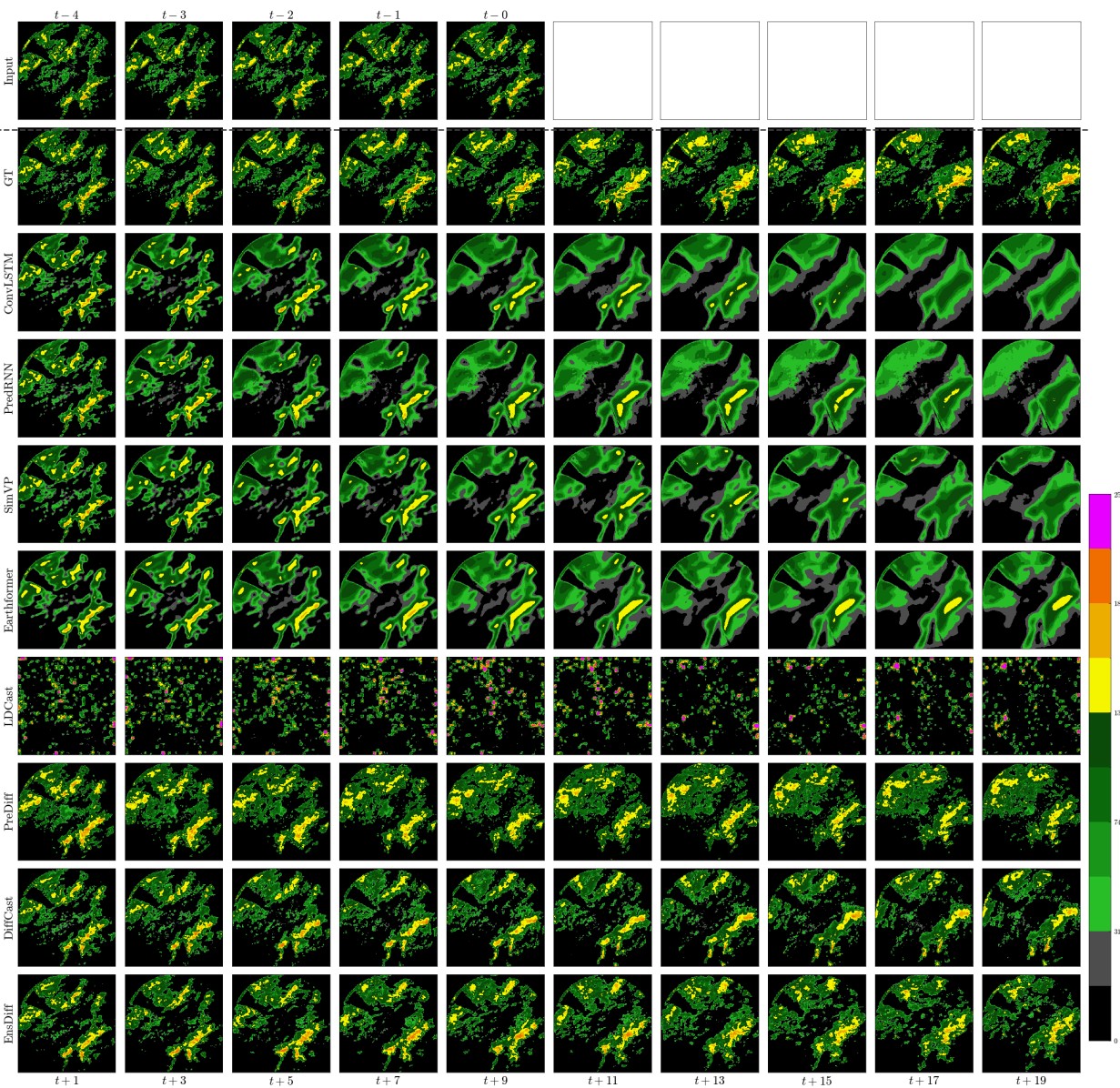

Figure 6: Visualization of all the models on the HKO-7 dataset (Sample 1)

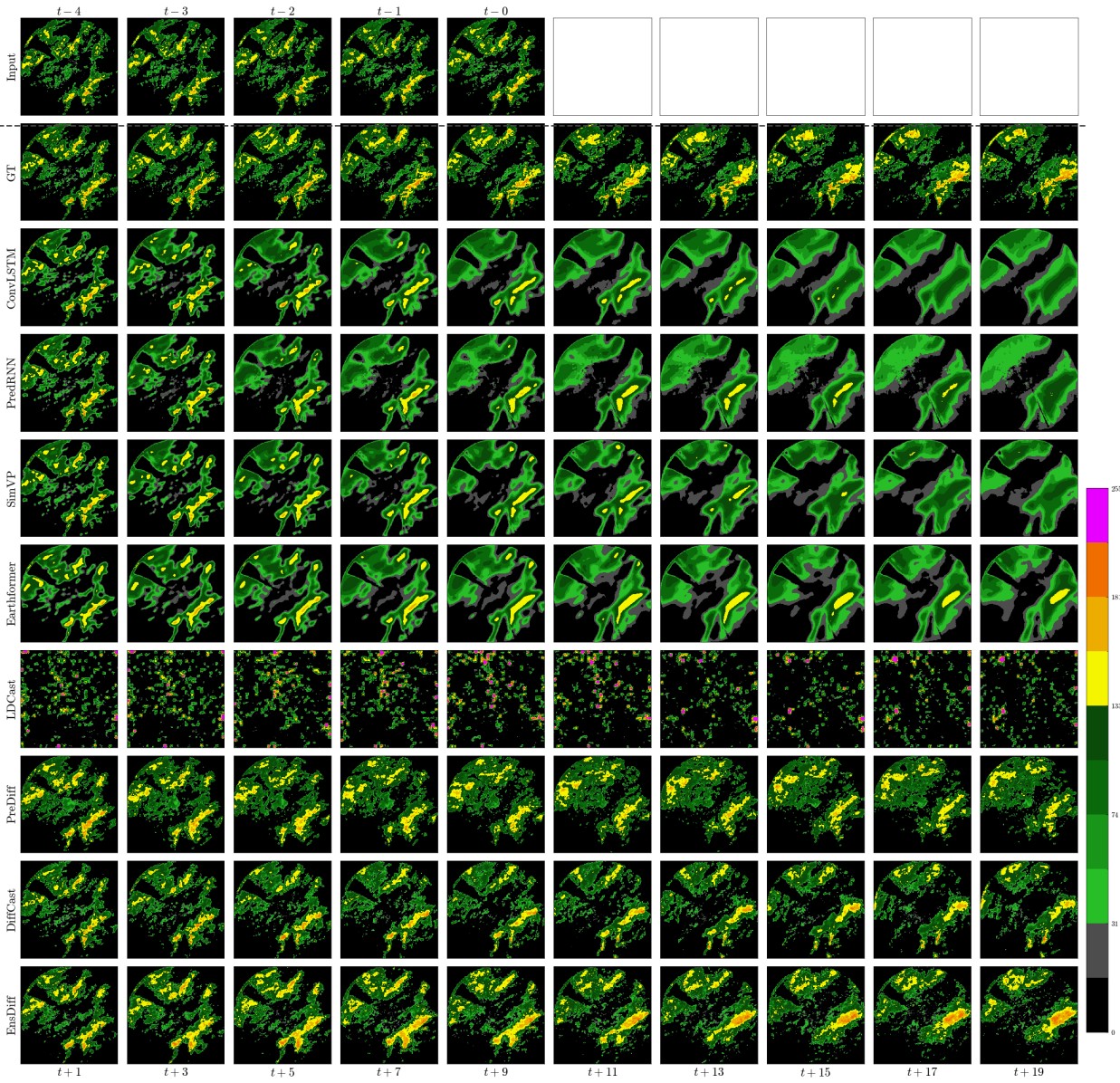

Figure 7: Visualization of all the models on the HKO-7 dataset (Sample 1)

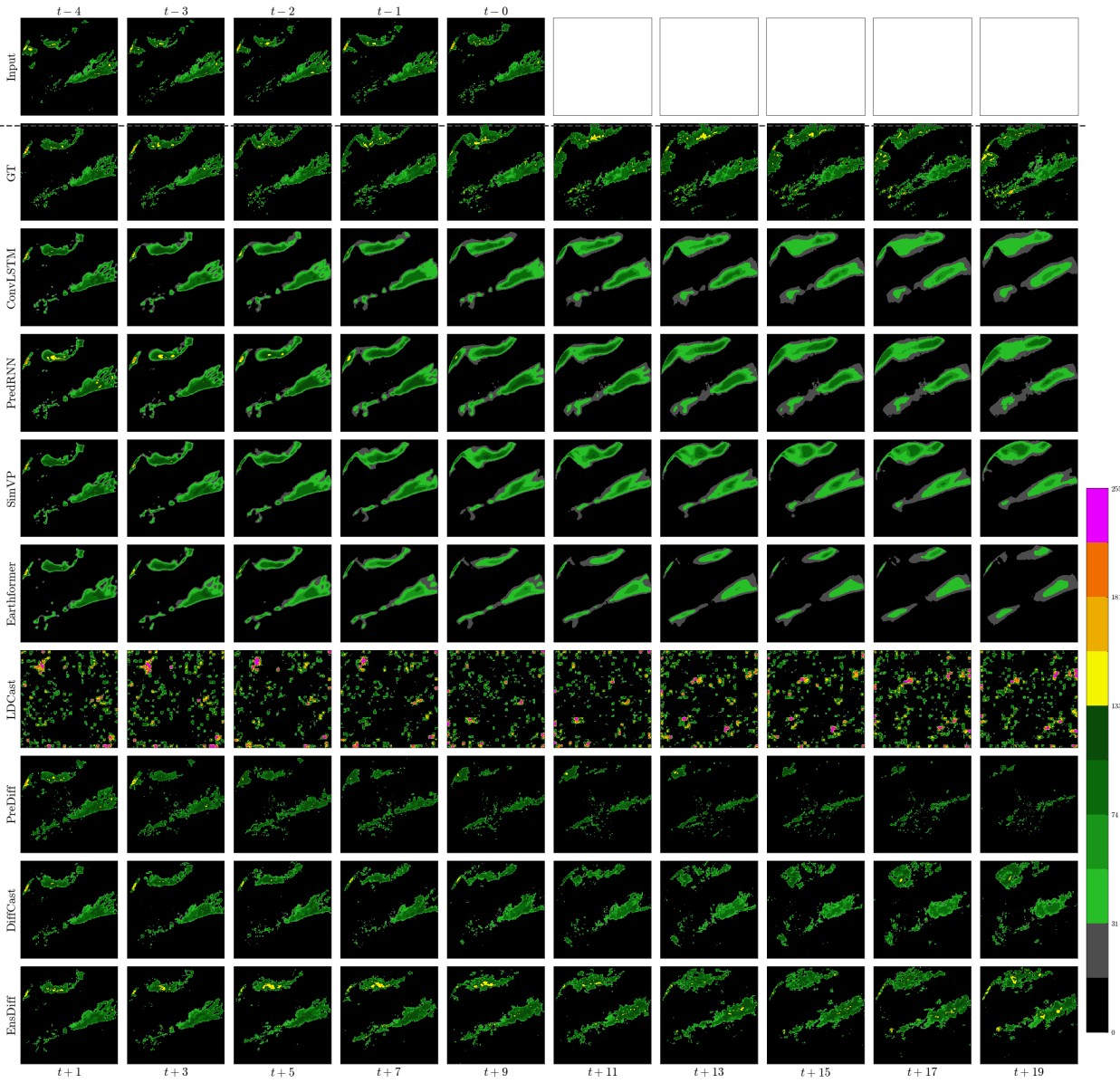

Figure 8: Visualization of all the models on the HKO-7 dataset (Sample 2)

## D    More Ablation Study Qualitative Results

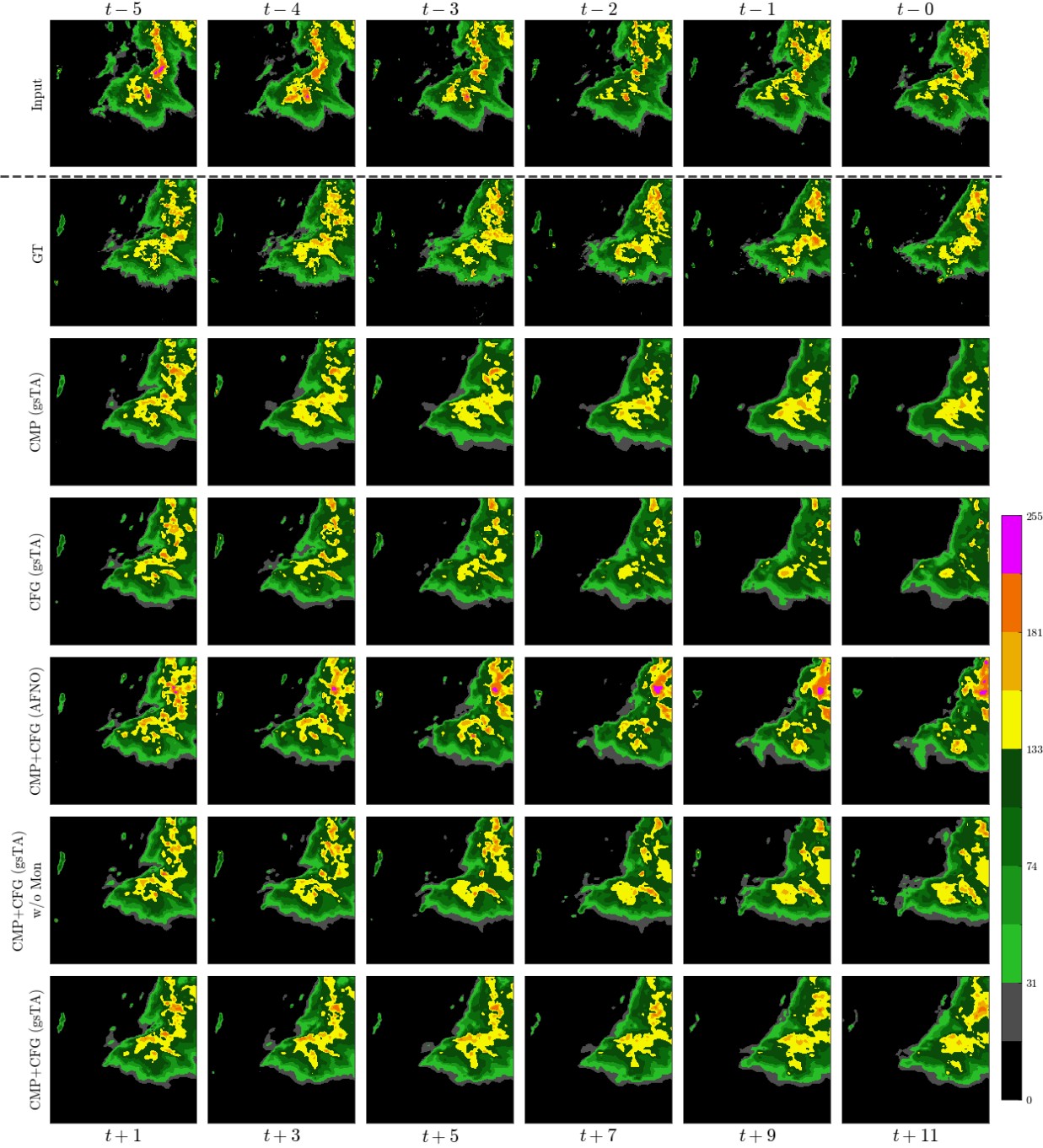

Figure 9: Visualization Sample 1 of the model variants in Ablation Study. (Mean Predictor) is indicated in the brackets. "w/o Mon" refers to without Monotonic Weighting, while it means with Monotonic Weighting for other without specified.

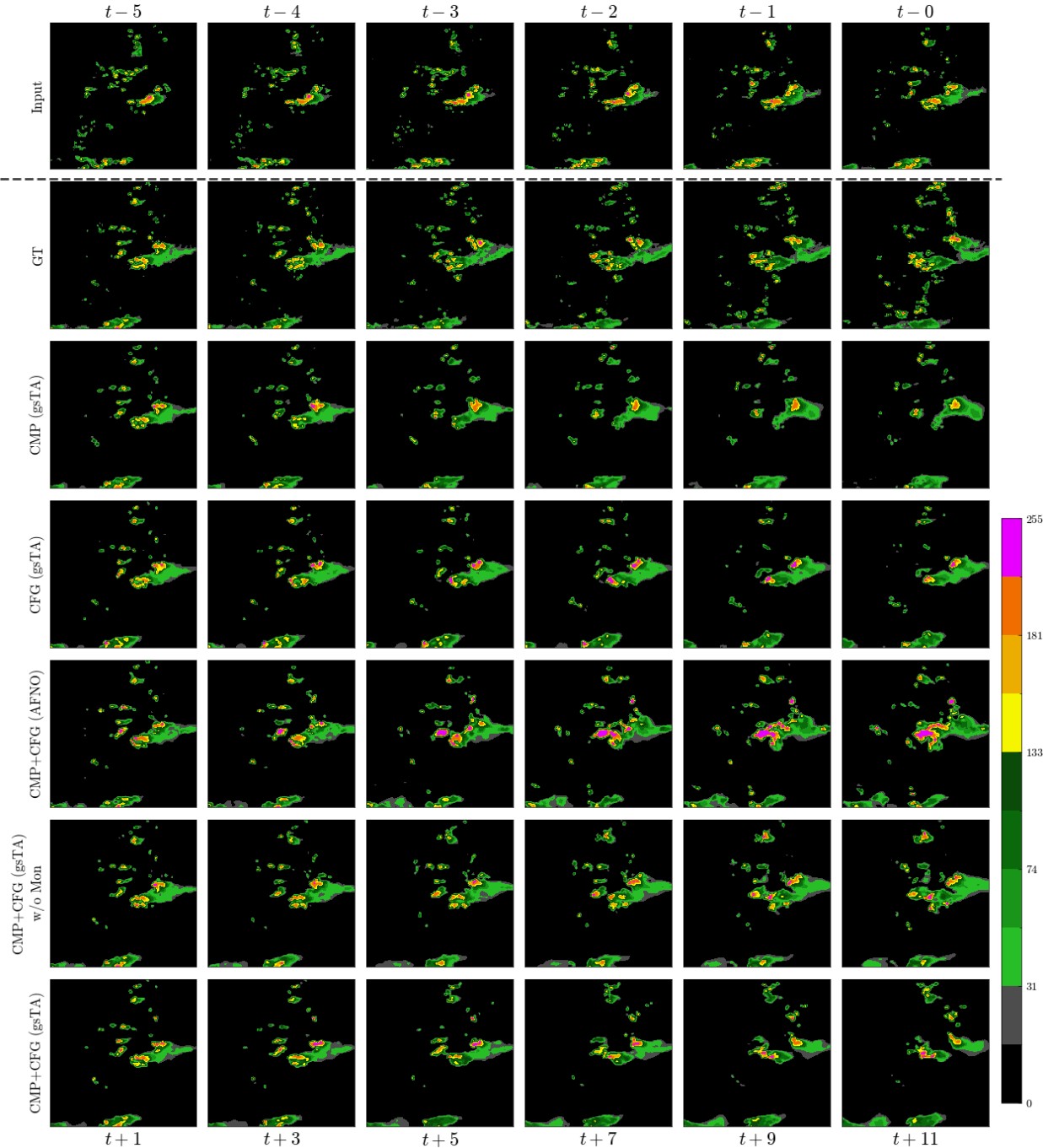

Figure 10: Visualization Sample 2 of the model variants in Ablation Study. (Mean Predictor) is indicated in the brackets. "w/o Mon" refers to without Monotonic Weighting, while it means with Monotonic Weighting for other without specified.

