# OpenReview forum: "EnsDiff: Ensemble Precipitation Nowcasting with Diffusion"
_TMLR — Withdrawn by Authors_

### Review · Reviewer_NP3h · 2025-07-14

**Summary Of Contributions:**

The paper focuses on using diffusion models for the Precipitation Nowcasting problem. The main contribution is the application of a continuous-time diffusion model to this problem, along with using a mean predictor to guide the sampling process. The evaluation on two datasets shows that the proposed method offers some advantages on certain metrics.

**Audience:**

Yes

**Broader Impact Concerns:**

No concerns.

**Claims And Evidence:**

No

**Requested Changes:**

From my point of view, the related works, method and experiments sections all need major revisions. See more detail at the Weakness section.

**Strengths And Weaknesses:**

### Strength

The Precipitation Nowcasting problem is indeed an important application and is a good test bed for spatio-temporal forecasting methods.

### Weakness

The biggest drawback of the paper is the readability. I had a hard time following the paper. Especially for the ones that didn't work on diffusion models and precipitation nowcasting problems like me.

The related works section to me is just a list of methods names. I only know the basics of diffusion models and GAN. But I don’t know how they are applied to the problem and how papers cited there are different from each other.

In the method section, the authors try to explain the key findings from the methods they apply. Most of them are direct citations of previous work and I think they should be well explained in the related work section. The current presentation makes it hard to separate this work’s contribution and previous ones.

I am also not sure about the significance of the results. The proposed method seems to have some advantage on the SEVIR dataset but not clearly on the HKO-7 dataset. Also not sure if it is common to evaluate the methods on 2 datasets in the Precipitation Nowcasting field, my guess is not. Also the implementation detail of baselines are missing.

### Questions

Sharpness is a property of a distribution and calibration is the alignment to real data. Shouldn’t the calibration be the ultimate goal? I know the point forecasting metrics and probabilistic forecasting metrics are two things, but a good method usually performs well on both (such as DiffCast on HKO-7 dataset).

In Section 2.1, why autoregressive methods with MSE loss produce blurry predictions? I don’t see the connection there. Are the predictions of different pixels independent? Isn’t it expected that neighbouring pixels have similar forecasts? Also why more blurry results show more uncertainty? The method should know nothing about uncertainty when optimizing MSE.

In Section 2.2.2, what is “mean of the ensemble from a diffusion model”? It means drawing samples from the diffusion model and compute the mean?

In Section 3.1, why does the prediction space have observed values? Its predicted values?

In Section 3.2.3, what does the author mean “sample quality over likelihood”? What is sample quality and how to measure it?

In Section 3.2.4, in inference time, how to get the residual?

---

### Review · Reviewer_C431 · 2025-07-14

**Summary Of Contributions:**

The focus of the paper is on probabilistic nowcasting of precipitation, with a downstream goal of preventing damage from extreme events such as flooding. In order to solve the nowcasting problem, the authors propose to learn a latent diffusion model. Novelties with regards to similar precipitation nowcasting methods include the usage of a mean predictor as a condition for classifier-free guidance of the diffusion model, a monotonic weighted diffusion objective and a continuous-time diffusion model. The evaluation on the SEVIR and HKO-7 datasets demonstrate competitive performance against a set of recent deterministic and probabilistic methods.

**Audience:**

Yes

**Broader Impact Concerns:**

None.

**Claims And Evidence:**

No

**Requested Changes:**

Major changes:

- The abstract seems to suggest that the proposed method overwhelmingly outperforms concurrent methods, while in fact state of the art results are only obtained for half of the test metrics. The description of the results in the abstract should be updated accordingly. A similar problem appears in the list of contributions.
- "Existing models utilizing deterministic and probabilistic frameworks have poor ensemble calibration (Pathak et al., 2024)." -> this statement is far too vague. Which tasks are you talking about? What does 'poor' mean? Are you saying that the ensemble calibration is poor for Pathak el al. only or for all existing models?
- "utilizing the equivalence of the Proper Scoring rule (Gneiting & Raftery, 2007) and the maximum likelihood estimation." -> what do the authors mean by "equivalence"? MLE is indeed a proper scoring rule, but not "the" proper scoring rule.
- the authors talk multiple times about the deterministic and probabilistic forecasting skills, without defining them, which is quite confusing since the "skill" of a probabilistic forecast often refers to its RMSE (see e.g. [1]).
- "Our model shortens the inference time" -> Compared to what? LDCast is faster according to appendix B. As mentioned in the experiment section, EnsDiff has much better performance than LDCast, so instead of saying that EnsDiff shortens the inference time you should say that it has a good tradeoff between performance and inference time, and possibly make a scatterplot of the different methods where the axes are inference time and CRPS. You could even train several versions of EnsDiff with varying performance/inference time compromises to draw a Pareto line.
- When describing existing deterministic approaches, the authors say that "the efficiency is low to run forward pass multiple times for the ensemble". This is questionable: if one has enough computational power then everything can be done in parallel with no overhead in computation time.
- Some previous works, such as PreDiff, are described in insufficient detail, which is a significant problem considering their proximity to the proposed method. The mention of a "UNet denoiser" is unclear. The authors should also better describe the models that are used as baselines, at least in an appendix section.
- "the prediction space is [F(y), y], where F(·) is the cumulative distribution function (CDF) of p(y|x), and y is the observed value." -> The notation "y" is used for 2 separate things : a random variable corresponding to a probabilistic forecast and a fixed observed value. You should use different notations to make this distinction appear explicitly.
- "Then, the distribution is said to be well calibrated when $Z_F$ ∼ Uniform(0, 1) is close to a uniform distribution" -> You are refering to PIT histograms, but again in a very unclear way, in part because of your ambiguous notations. You also do not mention the shortcomings of the PIT histogram which are described at length in the paper that you cite just before this.
- In equation (4) the indexes "t,...,1" seemingly appear out of nowhere
- the signal-to-noise ratio $\lambda$ is said to depend on the diffusion time t but this dependance should be made explicit by using the notation $\lambda_t$ (which the authors do only one time, with no explanation). One can also make a change of variable to make everything depend on $\lambda$ rather than on t but the authors do not explain this nor make use of it in their equations. This lack of clarity is detrimental to the understanding of the weighted diffusion objective. Also, why does $\lambda$ follow a Gaussian distribution with mean 2.4 and standard deviation 2.4?
- the simplification of equation (5) holds 2 major statements: an expression of L_T and the fact that other terms of equation (3) can be considered to be constant. Both of these statements should be separately and rigorously justified, either by proof or by refering to a paper that has this proof (which is not the case of the referred [2]).
- The developments around proper scoring rules are wrong. In the appendix, equation (14) makes no sense, and I suggest refering to section 2.2.1 of [3] for correcting it. Equation (17) is also wrong: instead of "a predictive distribution" $P_{\theta}$, the left side should involve the true distribution $P$ of $Y$, and the right-hand side a predictive distribution. Regarding the main text, it is not clearly explained that equation (6) is the criterion for a given scoring rule S to be proper. Equation (7) does not prove that negative ELBO is a proper scoring rule: to make such a proof, one would first need to make this negative ELBO depend on some predictive distribution rather than on only an observation y.
- The noising scheme of the diffusion model is not explicitly described, which is problematic because at some places you mention that Gaussian noise is added to the data, and then you say that at noise level $\sigma_{max}$ the distribution is Gaussian with mean 0, which means that the noise has replaced the original data rather than being added to it.
- The size of figure 2 makes it hard to read. The dashed line is also misplaced (which is the case for similar figures in the appendix too). I would recommend putting a similar figure with only a subset of methods in the main text, and put this comprehensive figure in the appendix instead.
- Considering the remark on the tradeoff between CSI and other metrics for the HKO-7 dataset, it would be interesting to have a figure showing the evolution of all evaluation metrics as a function of the number of training epochs.
- The ablation study is difficult to follow for several reasons. First, the authors dive into the analysis of the results before explaining what the different ablations consist in. They use acronyms that are only explicited in the legend of the table: they should be explicited in the main text as well. In the text, they give names to the ablation variants that do not directly correspond to the table, and they also do it in an ordering that does not match with this of the rows of the table. Regarding the comparison between CMP and CMP+CFG variants, it is very questionable to point to the fact that MAE and CRPS are approximately equal and then claim that individual frames are qualitatively better predicted by CMP+CFG, based on an example figure that might have been cherry-picked. Instead, the authors should focus on explaining the significant improvement of perceptual metrics brought by CFG.

Minor changes:

- The authors should make sure that all acronyms are defined before they are used, which is not the case for e.g. MSE, EDM and gSTA. The authors also inconsistently use the acronyms "gSTA" and "gsTA".
- It is claimed in the abstract that "Human experts can hardly make an appropriate decision with an ensemble forecast when forecast calibration and sharpness are not considered", which is not a very precise statement, since it is unclear what it means to consider the forecast calibration and sharpness.
- "runs ensemble prediction to obtain state and uncertainty estimation that leads to significant computational and energy costs" -> bad syntax
- "data-driven models that use abundant observation data to train can shorten the inference time of the model" -> the authors seem to say that data-driven models have a short inference time as a consequence of abundant training data, which is obviously not the point
- "For example, a forecast issues a 70% chance of rain. Then, it should rain approximately 70% of the time." -> the formulation is unnatural, I would suggest using a single sentence "if, ..., then..."
- "We categorize precipitation nowcasting models into mean predictor models and generative models. The former uses" -> the former category uses?
- "This shows the forecast uncertainty contributes" -> this shows that
- "sharpness corresponds to the concentration of the predictive distribution p(y|x), when it does not depend on observation y" -> what is this supposed to mean?
- "which forces the model not only reconstruct" -> not only to reconstruct
- "Thus, with updating the score, it decreases" -> bad formulation
- "our model is more preferrable to apply" -> bad formulation
- it should be explicitly said, either in table 2 or in its legend, that the horizontal lines between earthformer and LDCast separate the deterministic and the probabilistic methods
- "results comparable to those deterministic models" -> to the deterministic models
- "EnsDiff reconstruct the location" -> reconstructs
- "Then, the LDM is trained with a less noisy version in the latent. Then" -> repetion of "then"
- "for a sufficient of iterations" -> for a sufficient number
- There is an isolated "T" right before equation (18)

[1] Haynes, Katherine, et al. "Creating and evaluating uncertainty estimates with neural networks for environmental-science applications." Artificial Intelligence for the Earth Systems 2.2 (2023): 220061.

[2] Diederik Kingma, Tim Salimans, Ben Poole, and Jonathan Ho. Variational diffusion models. Advances in
neural information processing systems, 34:21696–21707, 2021

[3] Pacchiardi, Lorenzo, et al. "Probabilistic forecasting with generative networks via scoring rule minimization." Journal of Machine Learning Research 25.45 (2024): 1-64

**Strengths And Weaknesses:**

Strengths:
- The topic of precipitation nowcasting is of high interest, and it is especially relevant to treat it with an uncertainty-aware approach.
- The evaluation benchmark is rather complete, with an extensive set of recent and strong baselines as well as many evaluation metrics, enabling a good assessment of the performance of the proposed method.

Weaknesses:
- The writing is of poor quality, with numerous typographical errors and low clarity, which partially prevents from understanding what the authors are doing, what is novel and how significant the results are.
- While recent influencial related works are mentioned in the "related work" section, they are described with insufficient detail, which makes it difficult to fully understand what differs between the proposed method and the most closely related works.
- The mathematical developments suffer from inconsistent notations and insufficient textual explanation. Some of them are clearly wrong.
- There is not enough hindsight on the results: some claims are exagerated, and the authors mention no shortcomings or perspectives of improvement for their work.

---

### Review · Reviewer_FEcT · 2025-07-15

**Summary Of Contributions:**

The paper propose EnsDiff, a new model for precipitation nowcasting that uses diffusion models to generate ensemble predictions. The method balances forecast accuracy and uncertainty. It outperforms existing methods in both probabilistic (Continuous Ranked Probability Score) and deterministic metrics, as well as perceptual quality. The model achieves this by optimizing an Evidence Lower Bound (ELBO) objective through a monotonic weighted diffusion process.

EnsDiff also incorporates a mean predictor with an MSE constraint and utilizes classifier-free guidance to enhance deterministic skills and sample quality without sacrificing probabilistic performance. This approach leads to a more efficient and accurate model for operational weather prediction.

**Audience:**

Yes

**Broader Impact Concerns:**

The paper doesn't have ethical concerns.

**Claims And Evidence:**

Yes

**Requested Changes:**

The paper lacks description of important concepts and may not be self-contained for readers not in the field.
- For example, the authors should detail how metrics are calculated for example how Continuous Ranked Probability Score can be calculated from a diffusion model output.
- The EDM-monotonic weighting should be clearly displayed and discussed.

**Strengths And Weaknesses:**

Strength:
- the paper conducts careful ablation study.
- the paper shows good theoretical justification for the proposed EDM-monotonic weighting.
- the ablation study is thorough.

Weakness:
- the discussion on HKO-7 shows that the proposed model is sensitive to resolution, e.g., it failed to converge on 128 x 128 resolution but converged on 256 x 256. This shows potential instability in the training. It's unclear which design of the method leads to this problem

---

### Note · Authors · 2025-07-29

I have read and agree with the venue's withdrawal policy on behalf of myself and my co-authors.